# Association of Pre-S/S and Polymerase Mutations with Acute and Chronic Hepatitis B Virus Infections in Patients from Rio de Janeiro, Brazil

**DOI:** 10.3390/v14071375

**Published:** 2022-06-24

**Authors:** Camilla Rodrigues de Almeida Ribeiro, Katrini Guidolini Martinelli, Vinícius da Motta de Mello, Natália Spitz, Oscar Rafael Carmo Araújo, Lia Laura Lewis-Ximenez, Natalia Motta Araujo, Vanessa Salete de Paula

**Affiliations:** 1Laboratory of Molecular Virology, Oswaldo Cruz Institute, Oswaldo Cruz Foundation, Brasil Av., 4365 Manguinhos, Rio de Janeiro 21040-360, Brazil; camilla_almeida@hotmail.com (C.R.d.A.R.); nataliastd@gmail.com (N.S.); araujo.orc@gmail.com (O.R.C.A.); nmaraujo@ioc.fiocruz.br (N.M.A.); 2Department of Social Medicine, Espírito Santo Federal University, Espírito Santo 29075-910, Brazil; katrigm@gmail.com; 3Viral Hepatitis Laboratory, Oswaldo Cruz Institute, Oswaldo Cruz Foundation, Rio de Janeiro 21040-360, Brazil; vinicmk@hotmail.com (V.d.M.d.M.); lialewis.fiocruz@gmail.com (L.L.L.-X.)

**Keywords:** hepatitis B infection, acute, chronic, mutation

## Abstract

Several hepatitis B virus (HBV)-related factors, including the viral load, genotype, and genomic mutations, have been linked to the development of liver diseases. Therefore, in this study we aimed to investigate the influence of HBV genetic variability during acute and chronic infection phases. A real-time nested PCR was used to detect HBV DNA in all samples (acute, *n* = 22; chronic, *n* = 49). All samples were sequenced for phylogenetic and mutation analyses. Genotype A, sub-genotype A1, was the most common genotype in the study population. A total of 190 mutations were found in the pre-S/S gene area and the acute profile revealed a greater number of nucleotide mutations (*p* < 0.05). However, both profiles contained nucleotide mutations linked to immune escape and an increased risk of hepatocellular carcinomas (acute, A7T; chronic, A7Q). Furthermore, 17 amino acid substitutions were identified in the viral polymerase region, including the drug resistance mutations lamivudine and entecavir (rtL180M), with statistically significant differences between the mutant and wild type strains. Owing to the natural occurrence of these mutations, it is important to screen for resistance mutations before beginning therapy.

## 1. Introduction

Despite the availability of an efficient vaccine since the 1980s, an infection caused by the hepatitis B virus (HBV) is one of the main global public health problems. It is estimated that 2 billion people worldwide display evidence of a past or present HBV infection and 290 million people are chronic carriers [1,2]. The seroprevalence of HBV surface antigen (HBsAg) is age-specific and varies markedly by geographic region with the highest prevalence (>5%) in sub-Saharan Africa, East Asia, parts of the Balkan regions, the Pacific Islands, and the Amazon Basin of South America. A prevalence below 2% has been observed in regions such as Central Latin America, North America, and Western Europe [1]. Overall, almost half of the population of the world live in areas of high endemicity [2,3].

Brazil exhibits different patterns of endemicity of HBV infections, depending on the geographic region. The factors that may be responsible for these variations in prevalence are the demographic differences related to the epidemiological characteristics of the disease, sensitivity of laboratory techniques, fluctuations in viremia, and development of mutations that interfere with viral recognition by diagnostic tests [4].

Data from the Brazilian Ministry of Health reveal that between 1999 and 2017 218,257 confirmed cases of hepatitis B were reported in the country; of these, most were concentrated in the Southeast region (35.2%), followed by the South (31.6%), North (14.3%), Northeast (9.7%), and Central West (9.2%) regions [5].

The pre-S/S open reading frame (ORF), formed by pre-S1, pre-S2, and S regions, encodes three HBV surface proteins that make up HBsAg: L (large), M (middle), and S (small). HBsAg is the main envelope protein and includes regions involved in the binding of the virus to hepatocytes as well as the main epitopes recognized by neutralizing antibodies. Pre-S/S mutations can affect the antigenicity of HBsAg and have been shown to be responsible for false-negative results in several commercial tests for HBsAg, the evasion of anti-HBV immunoglobulin therapy, and the evasion of vaccine-induced immunity [6,7].

Molecular epidemiological studies have revealed remarkable differences in the geographic distribution of HBV genotypes and the frequency of mutations. Several naturally occurring HBV mutants, including those in the pre-S/S region, have clinical and epidemiological implications. HBV genotypes and mutations can play a critical role in viral pathogenesis, including changes in host immune recognition, increased virulence with increased viral replication, the facilitation of cell adhesion or penetration, and an association with hepatocarcinogenesis [6,7,8].

Viral and host factors as well as exogenous selection pressures typically define the predominant mutant species. Exogenous pressures include nucleoside/nucleotide analogs and interferon treatment as well as immune system intervention and vaccination [9].

In the clinical setting, variants are commonly selected during the immune-active disease phase when immune pressure is high or as a result of drug exposure. Owing to the overlapping nature of HBV ORFs, an in-frame mutation can affect the function of the protein encoded by that frame as well as the protein encoded by the overlapping reading frame [10].

In this study, we aimed to investigate the presence of mutations in the viral genome and to evaluate the association of these mutations in the pre-S/S and polymerase regions with cases of acute and chronic infections. The results presented herein identified several mutations related to an increased risk of hepatocellular carcinomas (HCCs), immune escape mutations, and drug resistance mutations. The frequency of these mutations and their distribution between the acute and chronic profiles are discussed and dated.

## 2. Materials and Methods

### 2.1. Ethics

The Oswaldo Cruz Institute/IOC/FIOCRUZ Research Ethics Committee approved this study (number CAE 06109812.4.0000.5248). All procedures were performed in accordance with the ethical standards of the responsible committee on human experimentation (institutional and national) and the Helsinki Declaration of 1975, as revised in 2008. All patients enrolled in the study signed an informed consent form after they were provided with all the necessary information to make an informed decision.

### 2.2. Study Population

The samples were randomly selected from a cohort of patients between 2014 and 2018 from the Viral Hepatitis Ambulatory, Viral Hepatitis Laboratory at the Oswaldo Cruz Foundation. This center receives suspected viral hepatitis patients, including acute and chronic cases, and their contacts.

A total of 71 individuals were included in this study. The patients were divided into two groups: Group I comprised 22 patients exhibiting symptoms of acute hepatitis B and group II comprised 49 patients with chronic hepatitis B (without antiviral therapy). To define the acute profile, the inclusion criteria were the presence of HBsAg, active viral replication indicator antigen (HBeAg), anti-HBc of the immunoglobulin M (IgM) class, and HBV DNA. To define the chronic profile, patients detected for HBsAg (>6 months) were selected. The liver biochemical parameters were measured in all patients. Patients with any other cause of liver damage (co-infection with other hepatotropic viruses, alcohol abuse, or autoimmune diseases) were excluded from the study.

### 2.3. Socio-Epidemiological Data Collection

The socio-epidemiological data, information about the infection, HBV treatment, and risk behaviors were obtained from the record of each patient or a questionnaire.

### 2.4. Biochemical Tests

The serum samples were subjected to biochemical doses of liver enzymes such as aspartate aminotransferase, alanine aminotransferase, alkaline phosphatase and gamma glutamyl transferase as well as total, direct, and indirect bilirubin through a system of quantitative determinations by photometry in a kinetic mode using a commercial kit (Labmax 560; Labtest, Brazil) according to the manufacturer’s instructions.

### 2.5. Detection of HBV Serological Markers in Serum Samples

All samples were tested for HBsAg (BioELISA HBsAg 3.0; BioKIT, Barcelona, Spain), anti-HBc IgM (BioELISA anti-HBc; BioKIT, Barcelona, Spain), HBeAg and anti-HBe (e411 Cobas; Roche Diagnostics, Basel, Switzerland), anti-HAV IgM (DiaSorin, Italy), anti-HCV (Murex anti-HCV 4.0; DiaSorin, Saluggia, Italy), anti-HEV (BioKIT, Barcelona, Spain), and anti-HIV (DS-EIA-HIVAGAB-SCREEN; RPC, Diagnostic System, Nijni Novgoro, Russia) according to the manufacturer’s instructions. Samples positive for other hepatitis or HIV were excluded from the study.

### 2.6. Molecular Assay and Phylogenetic Analysis

The HBV genetic material was extracted from HBV serum samples using a commercial kit (High Pure Viral Nucleic Acid Kit; Roche Diagnostics, Switzerland). The viral load of the HBV DNA was performed by a real-time PCR (qPCR) (TaqMan technology) using an Abbott Real-Time HBV Kit (Abbott Laboratories, Chicago, IL, USA); the amplification of the pre-S/S genomic region was performed by a nested PCR [11]. Amplicons were obtained from the nested PCR with an expected length of 1200 bp and were purified using a High Pure PCR Product Purification Kit (Roche Diagnostics, Switzerland) according to the manufacturer’s instructions. The pre-S/S sequences were determined from a single PCR fragment using a BigDye Terminator kit v3.1 (Applied Biosystems, Waltham, MA, USA) and the sequencing reactions were analyzed on an ABI3730xl automated sequencer (Applied Biosystems).

HBV genotyping was performed by a phylogenetic analysis of the pre-S/S gene with the reference sequences representing the HBV genotypes obtained from GenBank. The phylogenetic analysis was performed using the maximum likelihood method with an online version of the PhyML program [12]. The reliability of the phylogenies was estimated using the approximate likelihood ratio test based on the Shimodaira–Hasegawa-like procedure [13].

### 2.7. Analysis of Mutations

The presence of pre-S/S and drug resistance mutations was investigated using the Geno2pheno (HBV) online tool, an established web service in clinical use for analyzing HBV sequence data (http://hbv.geno2pheno.org/index.php accessed on 20 May 2021). The entire profile of the substitution in the nucleotides and amino acids analyzed by Geno2pheno (HBV) was evaluated and compared with the references specific for each genotype.

### 2.8. Data Analysis

The statistical analysis was performed using SPSS (version 15.0; SPSS Inc., Chicago, IL, USA). The descriptive statistics of the qualitative variables were determined by a frequency distribution and the quantitative variables were determined using the mean and standard deviation (SD). The normality of the data distribution was assessed using the Kolmogorov–Smirnov test. The association between the infection status and the personal and clinical characteristics was analyzed using the Pearson chi-squared test for the categorical variables and the ANOVA test for the continuous variables. Nucleotide mutations were stratified into synonymous and non-synonymous mutations; only non-synonymous mutations were considered in the analyses.

## 3. Results

Among the 71 patients who had their HBV DNA sequenced, 61.98% were male. The overall mean age was 42.45 ± 13.39 years with the men being almost 10 years older than the women (Table 1). We did not observe a statistically significant difference in age between the sexes in patients with acute infections. Patients with a chronic infection showed a significant difference in age between the sexes (*p* < 0.05). There was no significant difference in the viral load between the groups.

Of the total samples, 39 belonged to genotype A, sub-genotype A1, and 22 belonged to genotype A, sub-genotype A2. Sub-genotype A1 was imported from Africa and sub-genotype A2 was imported from Europe. One sample belonged to genotype D, sub-genotype D1; one belonged to genotype D, sub-genotype D3; and two belonged to genotype D, sub-genotype D4. Four samples belonged to genotype F, sub-genotype F1, and one sample belonged to genotype F, sub-genotype F2.

A total of 190 mutations were identified in the pre-S/S gene region: 53 nucleotide mutations, 53 amino acid (aa) mutations in the pre-S1 region, 26 aa mutations in the pre-S2 region, and 31 aa substitutions in the S region. In the reverse transcriptase (RT) domain, 17 aa substitutions were identified. Deletion mutations were not detected.

In the analysis of the nucleotide mutations in the pre-S/S region, 58.5% (31/53) mutations were found in the chronically infected individuals and 13 mutations were found exclusively in this group. We found that 41.5% (22/53) of mutations were present in greater numbers in the acute patients; of these, 4 were found exclusively in this group. C513T, T513A, and C666T mutations were more frequently identified in the chronic patients whereas T134C, C206A, C408T, T411C, and G625R mutations were more frequent in the acute patients. These mutations were significantly different between the two profiles (Table 2).

Stop codon mutations D42 *, C69 *, and W179 * were found more frequently in the acute patients and W182 * more frequently in the chronic patients; however, there was no significant difference between the two profiles for these mutations.

In the analysis of the aa substitution in the pre-S1 region, 43.4% (23/53) were more frequent in the chronic group, of which 12 were only found in this profile, and 56.6% (30/53) were more frequent in the acute group, of which 6 were only found in this profile. Mutations P41L, W43R, D47K, H51N, A62G, F63Y, Q100R, I108L, and I108V were more frequently noted in acute hepatitis whereas mutations H51T, Q104K, D114N, and D141E were more frequent in chronic hepatitis. These mutations exhibited significant differences between the two profiles (Table 3).

Among the mutations analyzed in the M region for pre-S/S, 57.7% (15/26) were more frequent in the chronic patients, of which 5 were only found in this profile. In the acute patients, 42.3% (11/26) were more frequent and no mutations exclusive to this profile were found. A7Q, Q13L, I42T, and D51G mutations were more frequently seen in the acute patients whereas A7T and A47S mutations were more frequent in the chronic patients; these exhibited significant differences between the two profiles (Table 4).

Among the mutations analyzed in the S region, 37.7% (21/31) were more frequent in the chronic patients with 8 only found in this profile; 32.3% (10/31) were more frequent in the acute patients with only 1 being found in this profile. F8L, G18V, F19C, V47G, and S61L mutations were more frequently observed in the acute patients whereas G44E and S45P mutations were more frequent in the chronic patients; these also showed significant differences between the two profiles (Table 5).

Only the patients with chronic infections possessed resistance mutations in the RT polymerase domain. Of the 49 patients with a chronic infection, 9 showed mutations in the RT polymerase domain and all of these belonged to genotype A. The rtL180M mutation showed a statistically significant difference (*p* < 0.05) between the mutant and the wild type. This mutation comprised a lamivudine and entecavir secondary resistance genotype. Regarding the distribution of the mutations in relation to the viral sub-genotypes (A1 or A2), no significant difference was identified (Table 6).

## 4. Discussion

Hepatitis B has a broad spectrum of manifestations, ranging from an acute, self-limiting illness with a resolution to cure, to evolving forms of multi-stage chronicity that can progressively culminate in liver cirrhosis and HCCs [14,15].

HBsAg is the major antigen of the viral envelope and comprises regions involved in viral binding to hepatocytes and the main epitopes recognized by neutralizing antibodies and T lymphocytes [16,17,18]. HBsAg contains a major hydrophilic region (MHR) and a cluster of B cell epitopes known as the “a” determinant, comprising amino acids 124–147 [19]. Mutations that cause a conformational change in the “a” determinant can affect the antigenicity of HBsAg, which is essential for inducing the production of protective antibodies, and are responsible for vaccine escape, escape from anti-HBV immunoglobulin therapy, and false-negative serological test results [17].

In the present study, we highlighted the importance of the nucleotide mutations T134C, G206A, C408T, T411C, and G625R in the acute patients and C513T, T513A, and C666T in the chronic patients in the pre-S/S region. Nucleotide mutations have also been observed in patients with cirrhosis and HCCs. Furthermore, these mutations lead to a high degree of quasi-species formations in patients with HBV infections, which is likely related to the severity of the infection [17,18,19].

The V47G mutation, related to the acute profile, and the F8L mutation, related to the chronic profile, are found outside the MHR in the S region, which may primarily affect T cell epitopes. These changes can also be considered to be naturally occurring immune escape mutations due to the host immune surveillance at the T cell level. Although no serious impacts of these mutations have yet been demonstrated, the proper reactivity of T-helper cells is a prerequisite for an adequate production of anti-HBs. The effective recognition of cytotoxic T lymphocytes is also required for the elimination of infected hepatocytes [17,18].

Q100R, I108L, and I108V (aa substitutions related to the acute profile) and Q104K, D114N, and D114E (aa substitutions related to the chronic profile) are mutations located in the MHR of HBsAg, in which the “a” determinant is located. Thus, these are important immune escape mutations that affect the antigenicity of HBsAg, essential for the induction of protective antibodies and responsible for escape from vaccine-induced immunity [17]. Similar mutations have been detected in immunocompromised patients and are thought to contribute to HBV reactivation in anti-HBs-positive individuals. This reactivation can lead to severe acute hepatitis, fulminant liver failure, and death [17,20].

We highlighted the P41L, W43R, P47K, H51N, A62G, and F63Y mutations in the acute phase and H51T in the chronic phase of the disease in the pre-S1 region. No significant association was found between the clinical status and the other mutations analyzed in this region. H51T, related to the chronic patient profile, has been associated with an increased risk of HCCs [21].

For the pre-S2 region, mutations A7Q, Q13L, I42T, and D51G were associated with the acute profile whereas A7T and A47S were associated with the chronic profile. Of these, we highlighted A7T and A7Q because a few studies have suggested that these mutations may be associated with an increased risk of HCCs [21]. For the S region, we highlighted the F8L, G18V, F19C, V47G, and S61L mutations in the acute phase and G44E and S45P in the chronic phase. No significant association was noted between the clinical status and the other mutations analyzed in these regions.

In the present study, several mutations related to the chronic profile were found to be associated with an increased risk of HCCs, suggesting that further research in the field of HBV genetic variability is necessary, especially to investigate the potential of a less favorable disease progression. Naturally occurring pre-S/S variants are frequently observed in patients with a chronic HBV infection and have been shown to influence liver disease progression. In this regard, pre-S/S variants should be routinely determined in HBV carriers to help identify those who may be at a greater risk of an unfavorable disease progression. Further studies are required to explore the molecular mechanisms of the pre-S/S variants involved in the pathogenesis of each disease stage [21].

In addition to the mutations mentioned above, D42 *, C69 *, and W179 * were identified more frequently in the acute patients and W182 * more frequently in the chronic patients although there were no statistically significant differences between the two profiles. Stop codon mutations in the pre-S/S region have been reported in patients with progressive liver disease [22]. However, the pathogenic effects of these naturally occurring mutations remain unknown. Stop mutations such as C69 * and W182 * have been identified in HCC tumors [23,24]. Functional studies of W182 * mutants have demonstrated greater cell proliferation and transformation abilities than those without the mutation [23]. All patients included in the study contained HBsAg. HBsAg is normally produced in infected individuals when stop codons are present in the pre-S1 or pre-S2 region; however, the large protein would not be translated if the stop codon was present in the pre-S1 region and the middle protein would not be translated if the stop codon was present in the pre-S2 region [25].

RT polymerase is encoded by the largest ORF in the genome. Owing to the lack of proofreading activity, it introduces random mutations into the HBV genome at a rate of approximately 10^−4^ to 10^−7^ mutations per site per year as a result of the highly error-prone nature of HBV RT [26,27].

The drugs most commonly used for the treatment of chronic hepatitis B (CHB) are immunomodulators, such as interferon-alpha and pegylated interferon-alpha, and nucleoside/nucleotide analogs, such as lamivudine, adefovir, entecavir, telbivudine, and tenofovir. However, drug resistance mutations often arise during the long-term use of therapies with a low barrier to resistance (such as lamivudine), leading to treatment failure and a progression to liver disease. For this reason, tenofovir and entecavir are preferable choices because of their high genetic barrier [28].

Primary drug resistance mutations are amino acid changes that cause a direct resistance to nucleoside/nucleotide analogs and decrease the viral susceptibility [29,30]. Secondary or compensatory mutations refer to amino acid substitutions that compensate for replication defects caused by primary drug resistance mutations and can reduce drug susceptibility by restoring the adequacy of the viral replication [30,31,32].

Of the 49 patients with a chronic infection, 9 (18.3%) exhibited mutations in the polymerase region and all of these belonged to genotype A. Of the 17 resistance mutations analyzed, the lamivudine and entecavir rtL180M resistance mutations showed a statistically significant difference (*p* < 0.05) between the mutant and the wild type. The rtL180M mutation is a secondary resistance mutation. Literature-based incidence data show that rtL180M has a higher natural incidence rate (2.96%) than other secondary mutations. In a study published by Zhang et al. [33], an overall rtL180M mutation frequency of 2.67% was reported. Other studies, including Fung et al. [34], Yamani et al. [32], and Mirandola et al. [35], reported that the prevalence rates of rtL180M were 10.0%, 2.08%, and 1.18% in Chinese, Indonesian, and Italian HBV carriers, respectively.

None of the patients included in this study had received an antiviral treatment. Reports on the incidence of pre-existing RT mutations in untreated patients are highly variable, ranging from 0% to 57% [35,36,37,38,39,40]. This large discrepancy between studies may be due to differences in factors such as patient geographic or ethnic origins, the sample size, and the viral genotypes. Several studies have reported a prevalence rate of pre-existing RT mutations (primary and secondary) of >5% in untreated patients [41].

A few studies have identified that the number of mutations in RT is associated with the progression of liver disease [42]. Zhu et al. [43] revealed that patients with multiple RT mutated sites demonstrated a significantly higher rate of liver fibrosis, suggesting a link between the viral mutations and the clinical progression of chronic hepatitis. Furthermore, a natural accumulation of RT mutations is involved in viral survival during chronic liver fibrosis.

The present study had a few limitations; the most notable was the small number of patients with other genotypes, which made it impossible to analyze the mutation profile related to the viral genotype.

## 5. Conclusions

In conclusion, we identified several mutations that may be associated with an increased risk of HCCs. Immune escape mutations distributed in both profiles were also observed in the chronic patients without antiviral treatment mutations in the polymerase region. Although the exact role of immune escape mutations in the pathogenesis of HBV reactivation is unknown, it is important to monitor these mutations in all patients with a history of HBV infections during the immunosuppression phase for prophylaxis in patients at risk of a reactivation. In addition, understanding the frequencies and clinical implications of the viral mutations can contribute to the improvement of diagnostic procedures, better planning of immunization programs, and creation of more efficient therapeutic protocols.

## Figures and Tables

**Table 1 viruses-14-01375-t001:** Demographic, epidemiological, clinical, and genotypic characteristics of the population.

	Total (*n* = 71)		Acute Infection (*n* = 22)		Chronic Infection (*n* = 49)		
Categorical Variables	*n*	%	*n*	%	*n*	%	
Gender							
Female	27	38.02	7	31.81	20	40.81	
Male	44	61.98	15	68.19	29	59.19	
Genotypes							-
A	61	85.91	18	81.81	43	87.75	
D	4	5.63	0	0.00	4	8.16	
E	1	1.40	0	0.00	1	2.04	
F	5	7.04	4	18.18	1	2.04	
			Acute infection		Chronic infection		
Continuous variables	mean	SD	mean	SD	mean	SD	*p*-Value *
Age (years)							
Female	36.78	12.17	41.00	10.55	35.30	12.59	
Male	45.93	13.02	41.87	10.82	48.03	13.73	
Viral load (log10 DNA IU/mL)	4.65	2.13	4.43	2.48	4.31	1.98	0.714

*n*: number of participants; SD: standard deviation; *: Student’s *t*-test.

**Table 2 viruses-14-01375-t002:** Analysis of nucleotide mutations in the pre-S/S region.

Mutations		Chronic Infection (*n* = 49)		Acute Infection (*n* = 22)		*p*-Value *
		*n*	%	*n*	%	
Pre-S/S						
nucleotides						
T134C	Wild type	49	100.0	18	81.8	*p* < 0.05
	Mutant	0	0.0	4	18.2	
G206A	Wild type	49	100.0	17	77.3	*p* < 0.05
	Mutant	0	0.0	5	22.7	
C408T	Wild type	48	98.0	18	81.8	*p* = 0.030
	Mutant	1	2.0	4	18.2	
T411C	Wild type	49	100.0	18	81.8	*p* < 0.05
	Mutant	0	0.0	4	18.2	
C513T	Wild type	38	77.6	22	100.0	*p* < 0.05
	Mutant	2	4.1	0	0.0	
T513A	Wild type	38	77.6	22	100.0	*p* < 0.05
	Mutant	9	18.4	0	0.0	
G625R	Wild type	19	38.8	10	45.5	*p* < 0.05
	Mutant	0	0.0	3	13.6	
C666T	Wild type	22	44.9	18	81.8	*p* = 0.004
	Mutant	27	55.1	4	18.2	

*n*: number of participants; *: chi-squared test (*p* < 0.05).

**Table 3 viruses-14-01375-t003:** Analysis of amino acid substitution in the pre-S1 region.

Mutations		Chronic Infection (*n* = 49)		Acute Infection (*n* = 22)		*p*-Value *
		*n*	%	*n*	%	
Pre-S1 region						
P41L	Wild type	46	93.9	16	72.7	*p* = 0.021
	Mutant	3	6.1	6	27.3	
W43R	Wild type	46	93.9	16	72.7	*p* = 0.021
	Mutant	3	6.1	6	27.3	
P47K	Wild type	49	100.0	18	81.8	*p* < 0.05
	Mutant	0	0.0	4	18.2	
H51T	Wild type	46	93.9	18	81.8	*p* < 0.05
	Mutant	2	4.1	0	0.0	
H51N	Wild type	46	93.9	18	81.8	*p* < 0.05
	Mutant	1	2.0	4	18.2	
A62G	Wild type	49	100.0	18	81.8	*p* < 0.05
	Mutant	0	0.0	4	18.2	
F63Y	Wild type	48	98.0	18	81.8	*p* = 0.030
	Mutant	1	2.0	4	18.2	

*n*: number of participants; *: chi-squared test (*p* < 0.05).

**Table 4 viruses-14-01375-t004:** Analysis of amino acid substitution in the pre-S2 region.

Mutations		Chronic Infection (*n* = 49)		Acute Infection (*n* = 22)		*p*-Value *
		*n*	%	*n*	%	
Pre-S2 region						
A7T	Wild type	43	87.8	18	81.8	*p* < 0.05
	Mutant	5	10.2	0	0.0	
A7Q	Wild type	43	87.8	18	81.8	*p* < 0.05
	Mutant	1	2.0	4	18.2	
Q13L	Wild type	48	98.0	18	81.8	*p* = 0.030
	Mutant	1	2.0	4	18.2	
I42T	Wild type	48	98.0	18	81.8	*p* = 0.030
	Mutant	1	2.0	4	18.2	
A47S	Wild type	16	32.7	13	59.1	*p* = 0.036
	Mutant	33	67.3	9	40.9	
D51G	Wild type	48	98.0	18	81.8	*p* = 0.030
	Mutant	1	2.0	4	18.2	

*n*: number of participants; *: chi-squared test (*p* < 0.05).

**Table 5 viruses-14-01375-t005:** Analysis of amino acid substitution in the S region.

Mutations		Chronic Infection (*n* = 49)		Acute Infection (*n* = 22)		*p*-Value *
		*n*	%	*n*	%	
S region						
F8L	Wild type	48	98.0	18	81.8	*p* = 0.030
	Mutant	1	2.0	4	18.2	
G18V	Wild type	48	98.0	18	81.8	*p* < 0.001
	Mutant	1	2.0	4	18.2	
F19C	Wild type	48	98.0	18	81.8	*p* = 0.030
	Mutant	1	2.0	4	18.2	
G44E	Wild type	41	83.7	22	100.0	*p* < 0.05
	Mutant	8	16.3	0	0.0	
S45P	Wild type	40	81.6	18	81.8	*p* < 0.05
	Mutant	6	12.2	0	0.0	
V47G	Wild type	48	98.0	18	81.8	*p* = 0.030
	Mutant	1	2.0	4	18.2	
S61L	Wild type	48	98.0	18	81.8	*p* = 0.030
	Mutant	1	2.0	4	18.2	
Q100R	Wild type	49	100.0	18	81.8	*p* < 0.05
	Mutant	0	0.0	4	18.2	
Q104K	Wild type	48	98.0	18	81.8	*p* = 0.030
	Mutant	1	2.0	4	18.2	
I108L	Wild type	44	89.8	18	81.8	*p* < 0.05
	Mutant	4	8.2	0	0.0	
I108V	Wild type	44	89.8	18	81.8	*p* < 0.05
	Mutant	1	2.0	4	18.2	
D114N	Wild type	41	83.7	22	100.0	*p* < 0.05
	Mutant	4	8.2	0	0.0	
D114E	Wild type	41	83.7	22	100.0	*p* < 0.05
	Mutant	4	8.2	0	0.0	

*n*: number of participants; *: chi-squared test (*p* < 0.05).

**Table 6 viruses-14-01375-t006:** Analysis of polymerase mutations.

Mutations		Chronic Infection (*n* = 49)		*p*-Value *
		*n*	%	
Mutations in the polymerase region				
rtV173E	Wild type	48	98.0	
	Mutant	1	2.0	
rtV173L	Wild type	48	98.0	
	Mutant	1	2.0	
rtL180M	Wild type	41	84.0	*p* < 0.05
	Mutant	8	16.0	
rtM204I	Wild type	46	94.0	
	Mutant	3	6.0	
rtM204V	Wild type	44	90.0	
	Mutant	5	10.0	
rtT184S	Wild type	48	98.0	
	Mutant	1	2.0	
rtM250A	Wild type	48	98.0	
	Mutant	1	2.0	
rtM250G	Wild type	48	98.0	
	Mutant	1	2.0	
rtM250Q	Wild type	48	98.0	
	Mutant	1	2.0	
rtM250P	Wild type	48	98.0	
	Mutant	1	2.0	
rtM250S	Wild type	48	98.0	
	Mutant	1	2.0	
rtM250T	Wild type	47	96.0	
	Mutant	2	4.0	

*n*: number of participants; *: chi-squared test (*p* < 0.05).

## Data Availability

The data that support the findings of this study are available upon request from the corresponding author. The data are not publicly available due to privacy or ethical restrictions.

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
