# Peer review of "Association of Pre-S/S and Polymerase Mutations with Acute and Chronic Hepatitis B Virus Infections in Patients from Rio de Janeiro, Brazil"

_viruses, 2022, doi:10.3390/v14071375_

Round 1
Reviewer 1 Report
After the revisions you made, P value column should be deleted from Table 1.
Author Response
June, 09 2022
Dear Editor
Viruses
Thank you for reviewing our article entitled “ASSOCIATION OF PRE-S/S AND POLYMERASE MUTATIONS WITH ACUTE AND CHRONIC HEPATITIS B VIRUS INFECTION IN PATIENTS FROM RIO DE JANEIRO, BRAZIL”. We are in full agreement with the observations made by the reviewers. We have revised, modified, and improved the manuscript accordingly. The modifications are highlighted in the manuscript, and their explanations are included below.
Thank you for your consideration.Sincerely,
Dr. De Paula, VS and Dr. Ribeiro, CRA
Reviewer 1
After the revisions you made, P value column should be deleted from Table 1.
P value column was deleted from Table 1.
Reviewer 2 Report
In the revisions, polymerase is spelled with an i. Please change to Y.
Author Response
June, 09 2022
Dear Editor
Viruses
Thank you for reviewing our article entitled “ASSOCIATION OF PRE-S/S AND POLYMERASE MUTATIONS WITH ACUTE AND CHRONIC HEPATITIS B VIRUS INFECTION IN PATIENTS FROM RIO DE JANEIRO, BRAZIL”. We are in full agreement with the observations made by the reviewers. We have revised, modified, and improved the manuscript accordingly. The modifications are highlighted in the manuscript, and their explanations are included below.
Thank you for your consideration.
Sincerely,
Dr. De Paula, VS and Dr. Ribeiro, CRA
Reviewer 2
In the revisions, polymerase is spelled with an i. Please change to Y.
Necessary corrections have been made.
This manuscript is a resubmission of an earlier submission. The following is a list of the peer review reports and author responses from that submission.
Round 1
Reviewer 1 Report
The manuscript by Ribeiro et al. reports sequencing the Pre-S/S region of acute and chronic HBV infections in Brazil, mostly of genotype A2.
In general English needs serious review, preferably by a native English speaker.
Abstract. Hepatitis B is capital B. aa are substitutions, not ‘mutations’.
HBsAg is surface, not envelope antigen.
In Methods, the quantification of viral load method is not described but data are reported.
Results
It should be clearly indicated that the identification of nt mutations and aa substitutions was done against the references specific to each genotype.
Nucleotide mutations should be stratified between synonymous and not synonymous, only the latter being relevant to analyses.
The presentation of mutations in Table 2 should be according to genes: Pre-S 1, Pre-S 2 and S gene each with its own enumeration.
Since 96% of sequences were of genotype A(2), the mutation and substitution analysis should concentrate on this genotype only. Mutations in the other 10 cases distributed between 3 genotypes should be excluded in order to avoid genotype biases.
In the S region, it should be pointed out that none of the substitutions listed occurred in the MHR region. In that context, it would be more interesting to replace this table 5 by a table showing the polymerase protein substitutions. As to the lamivudine/entecavir resistance mutation/substitution the alternative of infection with mutated variant or occurrence of such mutations spontaneously in a chronically infected patient should be discussed; either option possible in the chronic cases.
Discussion.
The substitutions in the MHR mentioned in the discussion (T134C, Q100R, I108L/V, Q104K, D114N/E) are not listed in Table 5. Why? They should be. In the literature, none of them is recognised as affecting HBsAg recognition.
This discussion is too long and several of the assertions regarding the impact of mutations are incorrect. The whole discussion should be re-considered once the cleaning and clarification of mutations and substitutions is done.
Author Response
May, 20 2022
Dear Editor
Viruses
Thank you for reviewing our article entitled “ASSOCIATION OF PRE-S/S AND POLYMERASE MUTATIONS WITH ACUTE AND CHRONIC HEPATITIS B VIRUS INFECTION IN PATIENTS FROM RIO DE JANEIRO, BRAZIL”. We are in full agreement with the observations made by the reviewers. We have revised, modified, and improved the manuscript accordingly. The modifications are highlighted in the manuscript, and their explanations are included below.
We hope that the manuscript is now acceptable for publication.
Thank you for your consideration.
Sincerely,
Dr. De Paula, VS and Dr. Ribeiro, CRA
Reviewer 1
The manuscript by Ribeiro et al. reports sequencing the Pre-S/S region of acute and chronic HBV infections in Brazil, mostly of genotype A2.
Regarding genotypes and sub-genotypes, of the total samples, 39 samples belonged to genotype A subgenotype A1 and 22 samples belonged to genotype A subgenotype A1. One sample belonged to genotype D sub-genotype D1, one belonged to genotype D sub-genotype D3, and two belonged to genotype D sub-genotype D4. Four samples belonged to genotype F sub-genotype F1, and one sample belonged to genotype F sub-genotype F2. Most of our samples belonged to genotype A, which was the most prevalent genotype in Brazil.
A highlighted paragraph with sub-genotype distributions has been added to the Results section on page 4.
In general English needs serious review, preferably by a native English speaker.
We agree and appreciate the suggestion, and the manuscript has been sent to a native speaker.
Abstract. Hepatitis B is capital B. aa are substitutions, not ‘mutations’.
Necessary corrections have been made to the Abstract and are highlighted in yellow.
HBsAg is surface, not envelope antigen.
Necessary corrections have been made to the Introduction and are highlighted in yellow.
In Methods, the quantification of viral load method is not described but data are reported.
Necessary corrections were made to the Methodology section. A highlighted paragraph with relevant information has been added to page 3.
HBV genetic material was extracted from HBV serum samples using a commercial kit (High Pure Viral Nucleic Acid Kit; Roche Diagnostics, Switzerland). Samples were tested for the presence of HBV DNA by real-time TaqMan PCR (qPCR) using the Abbott Real‐Time HBV Kit (Abbott Laboratories, Illinois, EUA), and amplification of the pre‐S/S genomic region was performed by nested PCR [11].
Results
It should be clearly indicated that the identification of nt mutations and aa substitutions was done against the references specific to each genotype.
We agree with and appreciate this suggestion. This information has been added to the methodology section and is marked in yellow.
Presence of Pre-S/S mutations and drug resistance mutations was investigated using the Geno2pheno [HBV] online tool, an established web-service in clinical use for analyzing HBV sequence data, at http://hbv.geno2pheno.org/index.php. The entire profile of mutations in nucleotides and amino acids analyzed by Geno2pheno [HBV] was evaluated and compared with references specific to each genotype.
Nucleotide mutations should be stratified between synonymous and not synonymous, only the latter being relevant to analyses.
Only non-synonymous mutations were considered in the mutation analysis. This information has been added to the Statistical Analysis section of the methodology.
Statistical analysis was performed using SPSS (version 15.0; SPSS Inc., Illinois, USA). Descriptive statistics of the qualitative variables were determined by frequency distribution and quantitative variables were determined using the mean and standard deviation. The normality of the data distribution was assessed using the Kolmogorov–Smirnov test. The association between infection status and personal and clinical characteristics was analyzed using the Pearson chi-square test for categorical variables and the ANOVA test for continuous variables. Nucleotide mutations were stratified into synonymous and non-synonymous mutations, and only non-synonymous mutations were considered in the analyses.
The presentation of mutations in Table 2 should be according to genes: Pre-S 1, Pre-S 2 and S gene each with its own enumeration.
We conducted an analysis of nucleotide mutations in the entire pre-S/S region; only the mutations that obtained a statistical difference were added to Table 2.
Since 96% of the sequences were of genotype A (2), mutation and substitution analysis should concentrate on this genotype only. Mutations in the other 10 cases distributed among the three genotypes were excluded in order to avoid genotype biases.
We appreciate this suggestion, but most of our samples belonged to genotype A sub-genotype A1; this information has been added to the results. We think it is important, despite genotype A being the most prevalent in Brazil, to know the proportion of circulating mutations of the different genotypes that are present in the country, since it is already evident that different genotypes and their associated mutations can influence different clinical outcomes.
In the S region, none of the substitutions listed occurred in the MHR region. In this context, it would be more interesting to replace Table 5 with a table showing polymerase protein substitutions. As for the lamivudine/entecavir resistance mutation/substitution, the alternative of infection with a mutated variant or occurrence of such mutations spontaneously in a chronically infected patient should be discussed; either option is possible in chronic cases.
The mutations Q100R, I108L, and I108V (aa mutation related to the acute profile) and Q104K, D114N, and D114E (aa mutation related to the chronic profile) were statistically significant in our study and are listed in Table 5. According to the literature, all of these mutations are located in the main hydrophilic region (MHR).
- Lazarevic, I.; Banko, A.; Miljanovic, D.; Cupic, M. Immune-Escape Hepatitis B Virus Mutations Associated with Viral Reactivation upon Immunosuppression. Viruses. 2019 24;11(9):778.
- Caligiuri, P.; Cerruti, R.; Icardi, G.; Bruzzone, B. Overview of hepatitis B virus mutations and their implications in the management of infection. World J Gastroenterol. 2016 7,22(1):145-54.
- Salpini, R.; Colagrossi, L.; Bellocchi, M.C.; Surdo, M.; Becker, C.; Alteri, C.; Aragri, M.; Ricciardi, A.; Armenia, D.; Pollicita, M.; Di Santo, F.; Carioti, L.; Louzoun, Y.; Mastroianni, C.M.; Lichtner, M.; Paoloni, M.; Esposito, M.; D'Amore, C.; Marrone, A.; Marignani, M.; Sarrecchia, C., Sarmati, L.; Andreoni, M.; Angelico, M.; Verheyen, J.; Perno, C.F.; Svicher, V. Hepatitis B surface antigen genetic elements critical for immune escape correlate with hepatitis B virus reactivation upon immunosuppression. Hepatology. 2015 61, 823-833.
Discussion.
The substitutions in the MHR mentioned in the discussion (T134C, Q100R, I108L/V, Q104K, D114N/E) are not listed in Table 5. Why? They should be. In the literature, none of them is recognised as affecting HBsAg recognition.
We appreciate this suggestion, and the mentioned mutations are now correctly allocated in Table 5. In the literature cited above, all these mutations are located in the main hydrophilic region (MHR) and could affect HBsAg recognition. The discussion has been reorganized for better understanding.
This discussion is too long and several of the assertions regarding the impact of mutations are incorrect. The whole discussion should be re-considered once the cleaning and clarification of mutations and substitutions is done.
We appreciate the suggestion. The discussion has been reordered.

Reviewer 2 Report
The manuscript fits the special issue of viral hepatitis in Brazil. However, major revisions should be made.
Abstract
Line 17: How many samples were positive?
Line 21: It is not clear what is 17 aa mutations. Please clarify.
Line 25: When treating with Tenofovir and Entecavir screening is not vital because of high genetic barrier. Please correct.
Introduction:
A description of the preS/S is missing (determinants and regions e.g. M region, determinant a and clinical/ diagnostic implications of the mutations).
Methods:
Please clarify why you did not the genotyping analysis of geno2pheno.
Results:
Table 1: The p value is not correctly located in the table. If the value if only of the chronic infection, please show it correctly in the table.
What about the patients? How many mutations did you find in any patient in both groups? Were there any differences in the mutations per person?
Please add a suitable table for all the categories with this data.
Correlation of the various mutations should be associated with the genotypes in the tables. Please add.
Did you find any deletion mutations? It should be mentioned in the text.
Discussion:
The second paragraph should appear in the introduction.
Line 219: Here, you do not highlight the importance of nucleotide mutations. You should highlight the association of the mutations to each clinical status.
Line 278: You should mention that today, Tenofovir and Entecavir are preferable because they have high genetic barrier. Thus, analysis is not always necessary before treatment. In addition, according to guidelines analysis is not mandatory.
Line 279: This paragraph can be removed.
Your patients had all HBsAg. However, some of them had stop codon mutations. Please deal with this conflict.
You should refer to your study limitations.
Conclusions:
Arrange the first sentence correctly.
Author Response
May, 20 2022
Dear Editor
Viruses
Thank you for reviewing our article entitled “ASSOCIATION OF PRE-S/S AND POLYMERASE MUTATIONS WITH ACUTE AND CHRONIC HEPATITIS B VIRUS INFECTION IN PATIENTS FROM RIO DE JANEIRO, BRAZIL”. We are in full agreement with the observations made by the reviewers. We have revised, modified, and improved the manuscript accordingly. The modifications are highlighted in the manuscript, and their explanations are included below.
We hope that the manuscript is now acceptable for publication.
Thank you for your consideration.
Sincerely,
Dr. De Paula, VS and Dr. Ribeiro, CRA
Reviewer 2
The manuscript fits the special issue of viral hepatitis in Brazil. However, major revisions should be made.
Abstract
Line 17: How many samples were positive?
All the samples included in the study were HBsAg-positive. The sentence has been replaced in the Abstract section to clarify this information. All changes have been highlighted in yellow in the revised manuscript.
Line 21: It is not clear what is 17 aa mutations. Please clarify.
Necessary corrections have been made to the Abstract and are highlighted in yellow.
Line 25: When treating with Tenofovir and Entecavir screening is not vital because of high genetic barrier. Please correct.
Necessary corrections have been made to the Abstract and are highlighted in yellow.
Introduction:
A description of the preS/S is missing (determinants and regions e.g. M region, determinant a and clinical/ diagnostic implications of the mutations).
A small paragraph with the requested information has been added to the Introduction. All changes have been highlighted in yellow in the revised manuscript.
The pre-S/S ORF, formed by the pre-S1, pre-S2, and S regions, encodes three HBV surface proteins that make up HBsAg: L (large), M (middle), and S (small). HBsAg is the main envelope protein and includes regions involved in binding of the virus to hepatocytes, as well as the main epitopes recognized by neutralizing antibodies. Pre-S/S mutations can affect the antigenicity of HBsAg and have been shown to be responsible for false-negative results in some commercial tests for HBsAg, evasion of anti-HBV immunoglobulin therapy, and evasion of vaccine-induced immunity [6,7].
Methods:
Please clarify why you did not the genotyping analysis of geno2pheno.
All samples were routinely genotyped as part of the diagnostic routine of the Viral Hepatitis Outpatient Clinic of Instituto Oswaldo Cruz, and phylogenetic analyses were conducted before the search for mutations in the geno2pheno software. Accordingly, we could confirm all the genotypes of our samples in the same manner.
Results:
Table 1: The p value is not correctly located in the table. If the value if only of the chronic infection, please show it correctly in the table.
Necessary corrections have been made to the table and are highlighted in yellow.
What about the patients? How many mutations did you find in any patient in both groups? Were there any differences in the mutations per person?
Please add a suitable table for all the categories with this data.
We did not conduct analyses for each patient, and the objective of this study was a global analysis of the main mutations found in our cohort of patients that would have a significant correlation with the different profiles of infection. We appreciate this suggestion and hope to be able to carry out the suggested analyses in future work.
Correlation of the various mutations should be associated with the genotypes in the tables. Please add.
Most of our patients belonged to genotype A, a genotype prevalent in Brazil, and owing to the low number of samples in the other genotypes, it was not possible to carry out an analysis of mutations by genotype.
Did you find any deletion mutations? It should be mentioned in the text.
The requested information has been added to the Results. All changes have been highlighted in yellow in the revised manuscript.
A total of 190 mutations in the pre-S/S gene region were found: 53 nucleotide mutations, 53 amino acid (aa) mutations in the pre-S1 region, 26 aa mutations in the pre-S2 region, and 31 aa mutations in the S region. In the reverse transcriptase domain, 17 aa mutations were identified. Deletion mutations were not detected.
Discussion:
The second paragraph should appear in the introduction.
A small paragraph with the requested information has been added to the Introduction. All changes have been highlighted in yellow in the revised manuscript.
Line 219: Here, you do not highlight the importance of nucleotide mutations. You should highlight the association of the mutations to each clinical status.
A small paragraph with the requested information has been added to the discussion section. All changes have been highlighted in yellow in the revised manuscript.
The present study highlights the importance of the nucleotide mutations T134C, G206A, C408T, T411C, and G625R in acute patients and C513T, T513A, and C666T in chronic patients in the pre-S/S region. Nucleotide mutations have also been observed in patients with cirrhosis and HCC. Furthermore, these mutations lead to a high degree of quasi-species formation in patients with HBV infection, which is probably related to the severity of the infection [17,18,19].
Line 278: You should mention that today, Tenofovir and Entecavir are preferable because they have high genetic barrier. Thus, analysis is not always necessary before treatment. In addition, according to guidelines analysis is not mandatory.
A small sentence containing the requested information has been added to the Discussion section. All changes have been highlighted in yellow in the revised manuscript.
The drugs most used in recent years for the treatment of CHB are immunomodulators, such as interferon-alpha and pegylated interferon-alpha, and nucleoside/nucleotide analogs, such as lamivudine, adefovir, entecavir, telbivudine, and tenofovir. However, drug-resistant mutations often arise during long-term use of therapies with a low barrier to resistance, such as lamivudine, leading to treatment failure and progression to liver disease. For this reason, Tenofovir and Entecavir are preferable because they have a high genetic barrier [28].
Line 279: This paragraph can be removed.
Necessary corrections have been made.
Your patients had all HBsAg. However, some of them had stop codon mutations. Please deal with this conflict.
The presence of HBsAg was the criterion used to include patients in the present study. Stop codon mutations were found more frequently in acute patients. All these patients progressed to cure, and no statistical difference was observed between the two profiles for these mutations. Conversely, some studies have shown that stop codon mutations in the pre-S/S region are found in patients with progressive liver disease. However, the pathogenic effects of these naturally occurring mutations remain unknown. Stop mutations such as C69* and W182* have been identified in HCC tumors. Functional studies of the W182* mutants demonstrated greater cell proliferation and transformation abilities than those without the mutation.
You should refer to your study limitations.
A small paragraph with the requested information has been added to the discussion section. All changes have been highlighted in yellow in the manuscript. The biggest limitation of this study was the small number of patients with other genotypes, which made it impossible to analyze the mutation profile related to the viral genotype.
Conclusions:
Arrange the first sentence correctly.
Necessary corrections have been made and highlighted in yellow.
In conclusion, we found several mutations that may be associated with an increased risk of HCC. Immune escape mutations distributed in both profiles were also observed in chronic patients without antiviral treatment mutations in the polymerase region. Although the exact role of immune escape mutations in the pathogenesis of HBV reactivation is unknown, it is important to monitor these mutations in all patients with a history of HBV infection during the immunosuppression phase for prophylaxis in patients at risk of reactivation. In addition, understanding the frequencies and clinical implications of viral mutations can contribute to the improvement of diagnostic procedures, better planning of immunization programs, and creation of more efficient therapeutic protocols.

Round 2
Reviewer 1 Report
Abstract. Amino acid substitutions, not mutations.
M&M. The requested description of viral load quantification was not done. What is described in the revision is qualitative.
There is no amino acid ‘mutations’ but ‘substitution’ when nt mutations are not synonymous.
Regarding subgenotype A1 and A2, it could be indicated that A1 was imported from Africa and A2 from Europe.
The recommended table displaying the aa substitution in the polymerase reading frame has not been included but should be. Substitution possibly interfering with anti-viral should be highlighted.
Author Response
May, 29 2022
Dear Editor
Viruses
Thank you for reviewing our article entitled “ASSOCIATION OF PRE-S/S AND POLYMERASE MUTATIONS WITH ACUTE AND CHRONIC HEPATITIS B VIRUS INFECTION IN PATIENTS FROM RIO DE JANEIRO, BRAZIL”. We are in full agreement with the observations made by the reviewers. We have revised, modified, and improved the manuscript accordingly. The modifications are highlighted in the manuscript, and their explanations are included below.
We hope that the manuscript is now acceptable for publication.
Thank you for your consideration.
Sincerely,
Dr. De Paula, VS and Dr. Ribeiro, CRA
Reviewer 1
Abstract. Amino acid substitutions, not mutations.
Necessary corrections have been made to the Abstract and are highlighted in yellow.
M&M. The requested description of viral load quantification was not done. What is described in the revision is qualitative.
Necessary corrections have been made to the Materials and Methods and are highlighted in yellow.
There is no amino acid ‘mutations’ but ‘substitution’ when nt mutations are not synonymous.
Necessary corrections have been made and are highlighted in yellow.
Regarding subgenotype A1 and A2, it could be indicated that A1 was imported from Africa and A2 from Europe.
The recommended information has been included in the manuscript and are highlighted in yellow.
The recommended table displaying the aa substitution in the polymerase reading frame has not been included but should be. Substitution possibly interfering with anti-viral should be highlighted.
The recommended table displaying the aa substitution in the polymerase reading frame has been included in the manuscript as table 6.

Reviewer 2 Report
Table 1: It is still not clear what the P value refers to. Between which variable or parameters of the chronic infection was the Chi-square test calculated? Please set up the table correctly.
Discussion: The explanation regarding the stop codons and the presence of the HBsAg should appear in the discussion, or at list mention shortly. You cannot ignore this conflict in the manuscript.
Author Response
May, 29 2022
Dear Editor
Viruses
Thank you for reviewing our article entitled “ASSOCIATION OF PRE-S/S AND POLYMERASE MUTATIONS WITH ACUTE AND CHRONIC HEPATITIS B VIRUS INFECTION IN PATIENTS FROM RIO DE JANEIRO, BRAZIL”. We are in full agreement with the observations made by the reviewers. We have revised, modified, and improved the manuscript accordingly. The modifications are highlighted in the manuscript, and their explanations are included below.
We hope that the manuscript is now acceptable for publication.
Thank you for your consideration.
Sincerely,
Dr. De Paula, VS and Dr. Ribeiro, CRA
Reviewer 2
Table 1: It is still not clear what the P value refers to. Between which variable or parameters of the chronic infection was the Chi-square test calculated? Please set up the table correctly.
The necessary information was included in the text above the table. All changes have been highlighted in yellow in the revised manuscript.
Discussion: The explanation regarding the stop codons and the presence of the HBsAg should appear in the discussion, or at list mention shortly. You cannot ignore this conflict in the manuscript.
A small paragraph was included in the discussion section. All changes have been highlighted in yellow in the revised manuscript.
